



# Reactive Organic Carbon Emissions from Volatile Chemical Products

Karl M. Seltzer[1], Elyse Pennington[2,3], Venkatesh Rao[4], Benjamin N. Murphy[5], Madeleine Strum[4], Kristin K. Isaacs[5], Havala O.T. Pye[5]

[1]Oak Ridge Institute for Science and Education Postdoctoral Fellow in the Office of Research and Development, US Environmental Protection Agency, Research Triangle Park, NC 27711
[2]Oak Ridge Institute for Science and Education Fellow in the Office of Research and Development, US Environmental Protection Agency, Research Triangle Park, NC 27711
[3]California Institute of Technology, Pasadena, CA 91125
[4]Office of Air and Radiation, US Environmental Protection Agency, Research Triangle Park, NC 27711
[5]Office of Research and Development, US Environmental Protection Agency, Research Triangle Park, NC 27711

*Correspondence to*: Havala O.T. Pye (Pye.Havala@epa.gov)

**Abstract.** Volatile chemical products (VCPs) are an increasingly important source of anthropogenic reactive organic carbon (ROC) emissions. Among these sources are everyday items, such as personal care products, general cleaners, architectural

coatings, pesticides, adhesives, and printing inks. Here, we develop VCPy, a new framework to model organic emissions from VCPs throughout the United States, including spatial allocation to regional and local scales. Evaporation of species in the VCPy framework is a function of the compound specific physiochemical properties that govern volatilization and the timescale relevant for product evaporation. We introduce the terms evaporation timescale and use timescale, respectively, to describe these processes. Using this framework, predicted national, per-capita organic emissions from VCPs are 9.7 kg person[-1] year[-1] (6.5 kgC

person[-1] year[-1]) for 2016, which translates to 3.12 Tg (2.10 TgC), making VCPs a dominant source of anthropogenic organic emissions in the United States. Uncertainty associated with this framework and sensitivity to select parameters were characterized through Monte Carlo analysis, resulting in a 95% confidence interval of national VCP emissions for 2016 of 2.68 – 3.60 Tg (1.81 – 2.42 TgC). This nationwide total is broadly consistent with the US EPA's 2017 National Emission Inventory (NEI); however, county-level and categorical estimates can differ substantially from NEI values. VCPy predicts larger VCP

emissions than the NEI for approximately half of all counties, with 5% of all counties featuring increases > 60%. Categorically, personal care products (150%) and paints/coatings (34%) feature the largest increases, whereas pesticides (-54%) and printing inks (-13%) feature the largest decreases. An observational evaluation indicates emissions of key species from VCPs are reproduced with high fidelity in the methods employed here (normalized mean bias of -13% with r = 0.95). Sector-wide, the effective secondary organic aerosol yield and maximum incremental reactivity of VCPs are 5.3% by mass and 1.59 g $O_3$ g[-1],

respectively, indicating VCPs are an important, and likely underrepresented to-date, source of secondary pollution in urban environments.

## 1 Introduction

Reactive organic carbon (ROC), which includes both non-methane organic gases and organic aerosol (OA), is central to atmospheric oxidant levels and modulates the concentration of all reactive species (Heald and Kroll, 2020; Safieddine et al., 2017). Gas-phase ROC features both biogenic and anthropogenic sources and, following oxidation, can lead to the formation of tropospheric ozone and secondary organic aerosol (SOA). Organic aerosol is often the dominant component of total fine particulate matter (PM$_{2.5}$) throughout the world (Jimenez et al., 2009; Zhang et al., 2007), and SOA is often the dominant



component of OA in both urban and rural settings (Jimenez et al., 2009; Volkamer et al., 2006; Williams et al., 2010; Xu et al., 2015). Since ozone and PM$_{2.5}$ are both associated with impacts on human health and welfare (U.S. Environmental Protection Agency, 2019a; U.S. Environmental Protection Agency, 2020) that are global in nature (Burnett et al., 2018; Mills et al., 2018) and persist at low concentrations (Di et al., 2017; Kazemiparkouhi et al., 2020), accurately understanding the sources, magnitude, and speciation of organic emissions is critical.


Historically, the dominant source of anthropogenic organic emissions in the United States has been motor vehicles (Khare and Gentner, 2018; McDonald et al., 2013; Pollack et al., 2013). However, successful emission reduction strategies implemented over several decades have dramatically reduced mobile emissions (Bishop and Stedman, 2008; Khare and Gentner, 2018; McDonald et al., 2013), resulting in substantial declines in both ambient gas-phase non-methane volatile organic compounds

(NMVOC) and OA concentrations (Gentner et al., 2017; McDonald et al., 2015; Pollack et al., 2013; Warneke et al., 2012). Due to these changes, volatile chemical products (VCPs) are now viewed as the foremost source of anthropogenic organic emissions (Khare and Gentner, 2018; McDonald et al., 2018). The U.S. EPA has long accounted for VCPs in the National Emissions Inventory (NEI) as the "solvent sector." In 1990, the mobile and VCP sectors were the two highest emitters of volatile organic compounds (VOC; a regulatory defined collection of organic species that excludes certain compounds, such as acetone) at the

national level. Mobile and VCP sources emitted 7.2 Tg and 5.0 Tg of VOCs, respectively (U.S. Environmental Protection Agency, 1995). By 2017, EPA estimates of VOC emissions from both the mobile and VCP sectors each dropped to 2.7 Tg (U.S. Environmental Protection Agency, 2020). For VCPs, factors driving the emissions decrease over this period include, but are not limited to, reformulation of consumer products (Ozone Transport Commission, 2016) and implementation of National Emissions Standards for Hazardous Air Pollutants regulations for industrial processes (Strum and Scheffe, 2016). Potentially complicating

the trend and assessment of relative roles of different sectors, new inventory methods have suggested that VCP emissions in the NEI could be biased low by a factor of 2-3 (McDonald et al., 2018).

The decades-long increasing relative contribution of VCPs to total anthropogenic organic emissions could have several important implications for modelling and improving air quality. First, modelling studies of SOA from anthropogenic VOCs have generally

focused on combustion sources (Hodzic et al., 2010; Jathar et al., 2017; Murphy et al., 2017), which are typically rich in aromatics and alkanes (Gentner et al., 2012; Lu et al., 2018). In contrast, emissions from VCPs occur through evaporation and contain large fractions of oxygenated species (e.g. glycol ethers, siloxanes), many of which feature uncertain SOA yields (McDonald et al., 2018). Second, adequate chemical mechanism surrogates for species common in VCPs (e.g. siloxanes) are lacking (Qin et al., 2020). As VCPs and their components could have significant SOA potential (Li et al., 2018; Shah et al.,

2020), revisiting VCP emissions mapping to chemical mechanisms could help reduce modelled bias, which has historically been difficult to resolve (Baker et al., 2015; Ensberg et al., 2014; Lu et al., 2020; Woody et al., 2016). Third, VCPs feature substantial quantities of intermediate-volatility organic carbon (IVOC) compounds (CARB, 2019) and better representing their source strength could help resolve the high IVOC concentrations observed in urban atmospheres (Lu et al., 2020; Zhao et al., 2014). Fourth, if the VCP sector is systematically low biased in the NEI or select urban areas, there could be implications for ozone

pollution (Zhu et al., 2019). Finally, reducing organic emissions from VCPs has traditionally been viewed through the lens of minimizing near-field chemical exposure (Isaacs et al., 2014) or mitigating ozone pollution (Ozone Transport Commission, 2018), both of which can be accomplished through product reformulation. For example, reducing the magnitude of regulatory VOC emissions from VCPs can be accomplished by reformulating a product with lower-volatility ingredients that are less likely to evaporate (Ozone Transport Commission, 2016). However, if these lower-volatility replacement ingredients eventually



evaporate on atmospherically relevant timescales, they could be efficient SOA precursors (Li et al., 2018).

Given these concerns, the need to understand and resolve differences among inventories becomes increasingly important. Here, we develop VCPy, a new framework to model organic emissions from VCPs throughout the United States, including spatial allocation to the county-level. In this framework, fate and transport assumptions regarding evaporation of a species in a product into ambient air are a function of the compound specific physiochemical properties that govern volatilization and the timescale
available for a product to evaporate. We introduce the terms evaporation timescale and use timescale, respectively, to describe these processes. Since product ingredients are considered individually, determination of emission composition is explicit. This approach also enables quantification of emission volatility distributions and the abundance of different compound classes. In addition, we test the sensitivity of predicted emission factors to uncertain parameters, such as use and evaporation timescales,
through Monte Carlo analysis, evaluate the VCPy inventory using published emission ratios, and estimate the effective SOA and ozone formation potential of both the complete sector and individual product use categories.

## 2 Methods

### 2.1 VCPy: A Framework for Estimating Reactive Organic Carbon Emissions from Volatile Chemical Products

The VCPy framework is based on the principle that the magnitude and speciation of organic emissions from VCPs are directly
related to (1) the mass of chemical products used, (2) the composition of these products, (3) the physiochemical properties of their constituents that govern volatilization, and (4) the timescale available for these constituents to evaporate (Fig. 1). VCPy attempts to address each of these points by utilizing the most relevant datasets available. Since the VCP sector includes residential, commercial, institutional, and industrial sources, a consistent stream of data sources for all product categories is difficult. As such, this work implements a hybridized methodology that utilizes the best features of prior emission inventory
methods, while introducing new methods to make improvements where necessary. The result produces national-level, per capita emission factors for all product categories in the VCP sector that can be further tailored for regional or localized analysis. The per capita basis is useful for comparison across frameworks and over time, but emissions can be recast in other units as needed.
Briefly, survey data are used to generate a 1st-order product composition profile for a composite of product types, which quantifies the fraction of organic, inorganic, and water components. The organics component is further divided into individual
species (e.g. ethanol, isobutane, isopropyl alcohol). A variety of data sources are used to estimate the national-level product usage and each composite is assigned a use timescale, reflecting the elapsed time between use and any explicit removal process. Finally, the characteristic evaporation timescale of each organic component is calculated using quantitative structure-activity relationship (QSAR) modelled physiochemical properties and compared to the assigned use timescale. If the characteristic evaporation timescale of the organic component is less than the assigned use timescale of the composite, it is assumed that the
compound is emitted. Else, the compound is retained in the product or other condensed phase (e.g. water) and permanently sequestered.

### 2.1.1 Product Use Categories (PUCs) and sub-Product Use Categories (sub-PUCs)

VCPy disaggregates the VCP sector into several components called Product Use Categories (PUCs). An individual PUC is not exclusively used in a singular setting (e.g. residential vs. commercial) and examples include Personal Care Products, Cleaning
Products, and Paints & Coatings. PUCs are further divided into sub-PUCs, which are composites of individual product types featuring similar use patterns. In addition to permitting tailored fate-and-transport assumptions, similar hierarchical product





schema are also useful for models estimating near-field exposure to chemicals, through routes such as dermal contact and indoor inhalation (Isaacs et al., 2020). As an example, there are two sub-PUCs allocated to the Personal Care Product PUC: Short Use Products and Daily Use Products. These two sub-PUCs are differentiated by the length of use prior to removal (i.e. the use

timescale). The mass of chemical products used and subsequent organic emission factors, which are the main output from VCPy, are calculated at the sub-PUC level (Fig. 1). Currently, there are nine PUCs and sixteen sub-PUCs implemented in VCPy (Table 1).

### 2.1.2 National-Level Product Usage

To estimate VCP product use, some prior work has used national economic statistics, such as market sales or shipment values

(e.g. U.S. Environmental Protection Agency, 2020; McDonald et al., 2018). Others have incorporated product usage statistics based on consumer habits and practices (e.g. Isaacs et al., 2014; Qin et al., 2020), but these statistics are generally unavailable for commercial and industrial chemical usage, which limits their application. To better ensure the capture of all chemical product usage, including usage in residential, commercial, institutional, and industrial settings, national economic statistics are utilized, where possible (Table S1).


Product usage from twelve sub-PUCs is estimated using national-level shipment statistics, commodity prices, and producer price indices. National-level economic statistics are retrieved from the U.S. Census Bureau's Annual Survey of Manufactures (ASM; U.S. Census Bureau, 2016a), which provides annual statistical estimates for all manufacturing establishments nationally. Values are available for all 6-digit North American Industry Classification System (NAICS) codes, provided as product shipment values

($ year$^{-1}$), and are reported with associated relative standard errors (generally < 5%). To translate shipment values ($ year$^{-1}$) to usage (kg year$^{-1}$), we use commodity prices ($ kg$^{-1}$) from the U.S. Department of Transportation's 2012 Commodity Flow Survey (U.S. Department of Transportation, 2015). An exception is for all Paint & Coating sub-PUCs. Commodity prices for these sub-PUCs are taken from the U.S. Census Bureau's Paint and Allied Products Survey (U.S. Census Bureau, 2011a) and representative of 2010. To translate these commodity prices, which are from 2010 and 2012, to values reflective of 2016, we use

producer price indices reported by the Federal Reserve Bank of St. Louis (U.S. Bureau of Labor Statistics, 2020). Commodity price indices from the Federal Reserve Bank are updated for all NAICS manufacturing codes monthly, which we average to create annual price indices (Table S2). An implicit assumption in this methodology is that manufacturing and product usage are, on average, annually balanced.

We preferentially utilize product usage numbers derived from the above methodology, when possible, as all data sources have the following characteristics: (1) they are nationally derived and therefore less influenced by regional differences in manufacturing and formulation, and (2) all datasets are freely available to the public. However, due to data limitations, product usage for four sub-PUCs are estimated using other sources. The Dry Cleaning and Oil & Gas product usage estimates are derived from the national-level solvent mass usage reported by an industry study (The Freedonia Group, 2016). The Miscellaneous

Products and Fuels & Lighter product usage estimates are derived from reported sales data, specific to California, from the California Air Resources Board's 2015 Consumer and Commercial Products Survey Data (CARB, 2019). These sales numbers are scaled upwards to a national-level by assuming equivalent per-capita product usage.



**2.1.3 1st-Order and Organic Product Composition**

Each sub-PUC features two composite profiles. The initial composite is the 1st-order product composition profile, which

disaggregates the total mass of each sub-PUC into its water, inorganic, and organic fractions (Table 2). Total organics are further decomposed into non-evaporative and evaporative organics. The quantification and accounting of evaporative organics in this framework are necessary as CARB's organic profiles are processed to exclude organics that are not anticipated to evaporate on atmospherically relevant timescales. For ten sub-PUCS, the 1st-order product composition profile uses data from the California Air Resources Board's 2015 Consumer and Commercial Products Survey (CARB, 2019). Various product types are sorted into

each sub-PUC and the 1st-order product composition profiles are calculated on a weighted basis using the reported sales from manufacturers and formulators in California. Due to omissions stemming from confidentiality concerns, not all sales and composition data from the survey are available. We utilize the publicly available portions of the data, which constitutes most of the survey and includes over 330 product types. For example, 126 product types and 20 product types were sorted into the General Cleaners and Adhesives & Sealants (Table S3) sub-PUCs, respectively.


For Architectural Coatings, Industrial Coatings, and Printing Inks, the 1st-order product composition profile is derived from data in the California Air Resources Board's 2005 Architectural Coatings Survey (CARB, 2007). The Architectural Coatings sub-PUC uses data from all profiles in the survey, which is dominated by flat paint, non-flat paints, and primers. Industrial Coatings and Printing Inks use the 1st-order product composition profiles of Industrial Maintenance coatings and Graphic Arts coatings,

respectively. The 1st-order product composition profile for aerosol coatings uses data from the California Air Resources Board's 2010 Aerosol Coatings Survey (CARB, 2012), which includes more than 20 aerosolized product types. Only the evaporative organic composition of these products was provided, so the remaining mass was evenly split between water and inorganics. For Dry Cleaning and Oil & Gas, as the product usage for these sub-PUCs were derived from the organic functional solvent mass usage, it is assumed that this mass is entirely evaporative organics.


The second composite is the organic composition profile. Again, the California Air Resources Board's 2015 Consumer and Commercial Products Survey (CARB, 2019) was used to derive the composition of organics for ten sub-PUCs (Table S4). Within each sub-PUC, all product types are mapped to an associated organic profile (CARB, 2018; see Table S3) and weighted based on their evaporative organic contributions to the total sub-PUC. For Architectural Coatings, an 88% water-based and 12%

solvent-based paint (CARB, 2007) composite is generated. Aerosol Coatings are calculated on a weighted basis using the potentially evaporative organic contributions reported by CARB's 2010 Aerosol Coatings Survey (CARB, 2012). The organic composition profiles for Industrial Coatings, Printing Inks, and Dry Cleaning all utilize profiles (3149, 2570, 2422, respectively) from EPA's SPECIATEv5.0 database (EPA, 2019b). Approximately 65% of the solvents used in the Oil & Gas sector are alcohols and the remainder are a broad range of hydrocarbons (The Freedonia Group, 2016). Since detailed composition data for

Oil & Gas solvents are sparse, all Oil & Gas alcohols are assumed to be methanol, as it is widely used in and emitted from Oil & Gas operations (Lyman et al., 2018; Stringfellow et al., 2017; Mansfield et al., 2018). The remaining 35% is allocated to naphtha, a blend of hydrocarbon solvents.

Several components within CARB profiles are lumped categories or complex mixtures. This includes naphtha, mineral spirits,

distillates, Stoddard Solvent, fragrances, volatile methyl siloxanes, and a series of architectural coating and consumer product "bins." All naphtha, mineral spirits, distillates, and Stoddard Solvent occurrences in individual profiles are treated as a single mineral spirits profile (Carter, 2015). Volatile methyl siloxanes include several compounds (e.g. $D_4$, $D_5$, $D_6$), all of which are



emitted in varying proportions (Janechek et al., 2017). Here, the lumped volatile methyl siloxane identity is preserved but the physiochemical properties of decamethylcyclopentasiloxane is applied to the surrogate. Fragrances are a diverse mixture of

organic compounds that include many terpenes and alkenes (Nazaroff and Weschler, 2004; Sarwar et al., 2004; Singer et al., 2006b). However, since the proportion of these constituents are unknown, all fragrances are physically treated as d-limonene since it is the most prevalent terpene emitted from fragranced products (Sarwar et al., 2004; Singer et al., 2006b). Finally, for the architectural coating and consumer product "bins," we use the representative chemical compositions derived by Carter, 2015.

### 2.1.4 Controls

There are two methods for controlling organic emissions from VCPs. The first method is through product reformulation, which would occur prior to product usage. Strategies that fit this definition include switching from a hydrocarbon solvent-based ingredient to one that is water-based, increasing the proportion of non-organics in a product, and reformulating a product with lower-volatility ingredients that are less likely to evaporate (Ozone Transport Commission, 2016). VCP emissions that stem from residential, commercial, and institutional settings rely on these pre-use controls to reduce emissions. Regulations often set VOC

content limits for chemical products (e.g. national standards: Section 183(e) of the Clean Air Act; 40 CFR 59), with California (e.g. CARB – Title 17 CCR) typically setting some of the most stringent limits in the country (Ozone Transport Commission, 2016). As the $1^{st}$-order and organic composition profiles utilized here are almost exclusively derived from product composition data, pre-use controls are implicitly represented. In fact, since the product composition data is from manufacturers and formulators in California, where product VOC content limits are typically more stringent than national regulations, applying

these profiles nationally likely results in conservative assumptions.

The second pathway of controlling organic emissions from VCPs is through post-use controls. Strategies that fit this definition include add-on controls, manufacturing process modifications, and disposal techniques. Add-on control strategies and manufacturing process modifications are limited to industrial and commercial emission sources, such as Industrial Coating (U.S.

EPA, 2007; U.S. EPA, 2008) and Printing Ink (U.S. EPA, 2006a; U.S. EPA, 2006b) facilities. Since adoption of these technologies vary widely in space and time, assigning a single post-use control efficiency is not considered. As several of these industrial sources (e.g. coatings, printing inks, dry cleaning) feature controls, as required by Section 112 of the Clean Air Act (40 CFR 63), this assumption could lead to localized high bias and will be refined in future work. Here, we only consider post-use controls through disposal techniques for the Oil & Gas and Fuels & Lighter sub-PUCs. For Oil & Gas, we assume that the

solvents used in these processes become entrained in the produced water at these sites. Since produced water is largely (~89-98%) reinjected for enhanced oil and gas recovery or disposal (Lyman et al., 2018; Liden et al., 2018), we apply a post-use control efficiency of 94% (i.e. average of reported reinjection rates) to this sub-PUC. However, it should be noted that reinjection frequency and solvent usage can vary regionally. For Fuels & Lighters, we assume 90% of the organics are destroyed through combustion upon use (CARB, 2019).

### 2.1.5 Evaporation and Use-Timescales

Fate-and-transport in the VCPy framework is a function of the compound specific evaporation timescale and the use timescale of each sub-PUC. It should be noted that this methodology explicitly results in the organic speciation of emissions differing from the organic composition of products from which they volatilize. For example, the composition of organics within a product may differ from the speciation of emitted organics if the product contains low-volatility compounds that do not evaporate on relevant

timescales.





The evaporation timescale is the compound specific (i.e. independent of the sub-PUC of interest), characteristic timescale of emission from a surface layer and is calculated using previously published methods (Khare and Gentner, 2018; Weschler and Nazaroff, 2008). This timescale is defined as a relationship between the mass of a compound applied and the rate of its emission,
which can be expressed by:

$$Evaporation\ Timescale\ [hr] = {M_{applied}}/{R_{emission}} = {K_{OA} \times d}/{v_e} \quad (1)$$

where $K_{OA}$ is the octanol-air partitioning coefficient of the compound, $d$ [m] is the assumed depth of the applied product layer,
and $v_e$ [m/hr] is the mass transfer coefficient of the compound from the surface layer into the bulk air, which is a function of aerodynamic and boundary layer resistances. Median values for $d$ [0.1 mm] and $v_e$ [30 m/hr] from Khare and Gentner (2018) are selected here. It should be noted that $v_e$ can vary substantially based on outdoor vs. indoor atmospheric conditions and future work will incorporate a two-box model to better account for such differences. A compound's $K_{OA}$ it is the ratio of an organic chemical's concentration in octanol to the organic chemical's concentration in air at equilibrium. It is often used to quantify the
partitioning behaviour of an organic compound between air and a matrix. As experimental values of $K_{OA}$ are sparse, modelled estimates from the quantitative structure-activity relationship (QSAR) model OPERA (Mansouri et al., 2018) are used here. All physiochemical properties, including OPERA results, are retrieved from the U.S. EPA's CompTox Chemistry Dashboard (https://comptox.epa.gov/dashboard; last access: August 31, 2020). While simple in setup, the assumptions adopted here broadly capture the relevant characteristic evaporation timescale for each compound.


Use timescale is the timescale available for a sub-PUC to evaporate and is based on the length of its direct use phase (i.e. the elapsed time between application and any explicit removal process). As this value is subjective, broad values are applied to each sub-PUC (Table S5). For example, it is assumed that all products used in the bath and shower are quickly sequestered and washed down the drain, thus largely unavailable for emission (Shin et al., 2015). As such, Short Use Products are assigned a
"Minutes" use timescale. In contrast, it is assumed that each person bathes once a day. Therefore, all Daily Use Products are assigned a "Days" use timescale.

Emissions are determined by comparing the calculated evaporation timescale for each component with the assigned use timescale for the sub-PUC. If the use timescale for the sub-PUC is greater than the evaporation timescale for a compound, the
compound is emitted. Else, the compound is retained in the product or other condensed phase and permanently sequestered. Overall, organic emissions (E) for the complete sector are calculated as a summation over all organic compounds, $i$, and sub-PUCs, $j$, as follows:

$$E = \sum_{i,j} \begin{cases} 0 & if\ Use\ Timescale_j < Evaporation\ Timescale_i \\ U_j \times f_{E_j} \times f_{S_{i,j}} \times \left(1 - f_{C_j}\right) & if\ Use\ Timescale_j \geq Evaporation\ Timescale_i \end{cases} \quad (2)$$


where $U$ is the product usage (Table 1), $f_E$ is the evaporative organic fraction (Table 2), $f_S$ is the fraction of an organic compound in the evaporative organics (Table S4), and $f_C$ is the fraction of emissions that feature post-use controls on a mass basis. Application of Eqn. 2 determines the difference between organic product composition and organic emissions speciation.





### 2.2 Uncertainty Analysis

The sensitivity of emission estimates to a variety of input variables are tested through a systematic Monte Carlo analysis. We perform 10,000 simulations where product usage, evaporative organic proportions, variables associated with the characteristic evaporation timescale, the assigned use timescale, and post-use control assumptions are tested, both individually and as a group. For product usage, the primary sources of uncertainty are shipment values provided by the ASM, commodity prices, the balance of imports (including tourism) and exports, and unused product disposal. The ASM provides standard error estimates for most

shipment values and are typically less than 5%. Uncertainty estimates are not provided for commodity prices and national-level exports generally outweigh traditional imports for most sub-PUCs (~2-15%; U.S. Census Bureau, 2016), but there are also imports of personal care products through tourism. Therefore, we conservatively assume there is a ±25% uncertainty (95% CI) to all product usage estimates. CARB does not provide uncertainty estimates associated with the composition of product types or sales proportions. To account for these uncertainties, as well as the uncertainties associated with generating composites, we

assume there is a ±25% uncertainty (95% CI) for all "Evaporative Organic" (Table 2) proportions. For the characteristic evaporation timescale, there are several layers of uncertainty. Application patterns vary by product type, which impacts assumptions regarding the depth of the chemical layer. In addition, indoor vs. outdoor product use and application of products to variable surface types (e.g. absorbing vs. non-absorbing) can impact mass transfer rates. As such, we apply broad uncertainties for variables associated with the characteristic evaporation timescale. We assume $d$ (i.e. the depth of the applied chemical layer)

is lognormally distributed with a median value of 0.1 mm (95% CI ~ [0.01 mm – 1 mm]) and $v_e$ (i.e. the mass transfer coefficient) is normally distributed with a mean value of 30 m/hr (95% CI = [10 m/hr – 50 m/hr]). Since use timescales are categorical (e.g. minutes, days, years), we apply uncertainty by assuming the 95% CI of the assigned use timescale features a ±1 categorical uncertainty (e.g. mean: minutes; 95% CI = [seconds – hours]). Finally, for non-zero, post-use controls, we assume a ±25% uncertainty (95% CI). Additional avenues of uncertainty may persist but are difficult to quantify and therefore not

included here. For example, due to the scarcity of large-scale product surveys, many of the 1st-order product composition profiles (e.g. Architectural Coatings) and organic profiles (e.g. Printing Inks) used in this analysis are more than a decade old. As a result, the proportion of organics in these product types and their organic components may have changed in the interim period.

### 2.3 Spatial Allocation of National-Level Emissions

Emissions are calculated at the national-level and spatially allocated to the county-level using several proxies. Ten sub-PUCs,

including all Cleaning Products and Personal Care Products, are allocated using population (Table S6; U.S. Census Bureau, 2020). Four sub-PUCs (Industrial Coatings, Allied Paint Products, Printing Inks, Dry Cleaning), all typically industrial in nature, are allocated using county-level employment statistics from the U.S. Census Bureau's County Business Patterns (U.S. Census Bureau, 2018). The employment mapping scheme for these four sub-PUCs utilize the methods from the 2017 NEI (U.S. EPA, 2020). On occasion, data in the County Business Patterns (CBP) is withheld due to confidentiality concerns. In those instances,

we take the mid-point of the range associated with each data suppression flag. For Agricultural Pesticides, emissions are allocated based on county-level agricultural pesticide use and again taken from the 2017 NEI (U.S. EPA, 2020). Oil & Gas emissions are allocated using oil and gas well counts (U.S. EIA, 2019).

### 2.4 Inventory Evaluation

Previously published emission ratios from the Los Angeles basin during the summer of 2010 (de Gouw et al., 2018; de Gouw et

al., 2017) are used to evaluate the VCPy emissions inventory (Table S7). Emissions ratios are generated by post-processing observed concentrations of organic gases, typically normalized to carbon monoxide (CO) or acetylene, to a period of "no





chemistry" (Borbon et al., 2013; de Gouw et al., 2005; Warneke et al., 2007). As the air parcel is not photochemically aged (i.e. "no chemistry"), it is an ideal tool for evaluating an emissions inventory. An important caveat is that this method assumes the species being used for normalization (e.g. CO) is accurately inventoried and measured.


Since the emission ratios are not specific to a sector and represent total emissions, all other sectors must be quantified and speciated. For this purpose, all non-VCP anthropogenic emissions from the 2017 NEI (U.S. EPA, 2020) are collected and speciated using EPA's SPECIATEv5.0 database (EPA, 2019b; Table S8). This includes all on road, nonroad, nonpoint, and point sources. All VCP emission from the 2017 NEI are also collected and speciated for supplementary evaluation. In addition,

biogenic emissions of ethanol, methanol, and acetone for May and June of 2016, as simulated by the Biogenic Emission Inventory System (Bash et al., 2016), were included to capture non-anthropogenic sources of these compounds. May and June were selected to coincide with the observational sampling months (de Gouw et al., 2018; de Gouw et al., 2017). As the observed emission ratios are specific to the Los Angeles basin, we derive all VCPy inventory emission ratios using data for Los Angeles County. Total CO emissions, including all on-road, non-road, non-point, and point sources, for Los Angeles County in 2017 are

~320 Gg. While the observed and VCPy inventory emission ratios are separated by 6-7 years, the ambient non-methane hydrocarbon to CO concentration ratio in Los Angeles has been consistent for several decades, indicating changes in emission controls feature similar improvements for both pollutants over time (McDonald et al., 2013). In addition, the magnitude of observed emission ratios for a given region do not appreciably change over marginal time horizons (Warneke et al., 2007).

## 2.5 Air Quality Impact Potential

Each organic compound is assigned a SOA yield and Maximum Incremental Reactivity (MIR) to facilitate an approximation of the potential air quality impacts of VCPs. For SOA, a wide collection of published yields, including both chamber results and prediction tools, were utilized (Fig. S1). These include: (1) all linear alkanes use a quadratic polynomial fit to the volatility basis set (VBS) data from Presto et al., 2010 at 10 µg/m³; (2) all cyclic alkanes use linear alkane yields that are three carbons larger in size (Tkacik et al., 2012); (3) all branched alkanes use yields obtained from the Statistical Oxidation Model (SOM; Cappa and

Wilson, 2012), as reported in McDonald et al. (2018); (4) benzene and xylenes use the average yields from Ng et al., 2007 under high-$NO_x$ conditions; (5) toluene uses the average from Ng et al., 2007 under high-$NO_x$ conditions and the VBS data from Hildebrant et al., 2009 at 10 µg/m³; (6) all alkenes use yields obtained from SOM, as reported in McDonald et al. (2018); (7) volatile methyl siloxanes use the two-product model parameters from Janecheck et al., 2019, which includes additional SOA yields from Wu and Johnson 2017, at 10 µg/m³; (8) all glycol ethers use chamber results and molecular structure relationships

from Li and Cocker 2018 for reported and unreported glycol ethers, respectively; (9) benzyl alcohol uses the average of the lower bound yields reported by Charan et al., 2020; (10) all remaining non-cyclic oxygenates, where available, use the arithmetic average of SOM results and a 1-D VBS approach, as reported by McDonald et al., 2018; (11) all remaining cyclic oxygenates, where available, use yields obtained from SOM, as reported by McDonald et al., 2018; (12) all halocarbons and compounds with less than five carbons are assigned a yield of zero; and (13) all remaining species are conservatively assigned a yield of zero if

the effective saturation concentration (i.e. $C^* = (P^{vap} \times MW)/(R \times T)$) is $\geq 3 \times 10^6$ µg/m³ and assigned the same yield as n-dodecane if the effective saturation concentration is $< 3 \times 10^6$ µg/m³. The MIR of each compound, which measures the formation potential of ozone under various atmospheric conditions where ozone is sensitive to changes in organic compounds (Carter, 2010b), is calculated using the SAPRC-07 chemical mechanism (Carter, 2010a) and expressed as a mass of additional ozone formed per mass of organic emitted (Carter, 2010b).





**3 Results and Discussion**

**3.1 National-Level PUC and sub-PUC Emissions**

National-level, per-capita organic emissions from VCPs are 9.7 kg person$^{-1}$ year$^{-1}$ (6.5 kgC person$^{-1}$ year$^{-1}$) for 2016 (Table 3), which translates to 3.12 Tg (2.10 TgC). When filtered to remove regulatory exempt organics, total emissions from VCPs are 2.6 Tg of VOC. In comparison, the 2017 NEI reports a combined total of 2.6 Tg of VOC emissions for on-road mobile, non-road

mobile, and other mobile (i.e. aircraft, commercial marine vessels, and locomotives) sources, respectively. Therefore, when measured as VOC, the VCP sector is equal in magnitude to the sum of all mobile sources nationally, which is broadly consistent with the national-level emissions estimate from the 2017 NEI. Categorically, emission factors are largest for Paints & Coatings, which total 3.4 kg person$^{-1}$ year$^{-1}$ (2.3 kgC person$^{-1}$ year$^{-1}$) and are approximately 35% of the total sector (Table 3). The next largest PUCs are Personal Care Products and Cleaning Products, which contribute 2.1 kg person$^{-1}$ year$^{-1}$ (21%) and 2.0 kg

person$^{-1}$ year$^{-1}$ (20%), respectively. Printing Inks, Adhesives & Sealants, and Pesticides each account for 6-8% each, and the remaining PUCs contribute less than 2% in total.

For the complete sector (Fig. 2), the most abundantly emitted compound class are oxygenated species (53%), followed by alkanes (31%; including straight-chained, branched, and cyclic), aromatics (8%), alkenes (5%), and halocarbons (3%).

Individually, organic emissions are dominated by ethanol (Daily Use Products, General Cleaners), acetone (Paints & Coatings, General Cleaners), isopropyl alcohol (Daily Use Products, General Cleaners), toluene (Paints & Coatings, Adhesives & Sealants), n-tetradecane (Printing Inks), fragrances (Daily Use Products, General Cleaners), propane (Aerosol Coatings, Industrial Coatings), and volatile methyl siloxanes (Daily Use Products, Adhesives & Sealants). Each of these species comprise > 3% of total VCP organic emissions.


In terms of volatility classification (Donahue et al., 2012), as determined by the effective saturation concentration (i.e. $C^*$), total emissions are predominately VOCs ($C^* > 3 \times 10^6$ µg $m^{-3}$), but there are also considerable contributions from IVOCs ($3 \times 10^2$ µg $m^{-3} < C^* < 3 \times 10^6$ µg $m^{-3}$; Fig. 2-3). IVOC emissions, which are efficient SOA precursors (Chan et al., 2009; Presto et al., 2010), are approximately 20% of total emissions. Of this 20% that are IVOCs, 55% are oxygenated compounds

(mainly Texanol™, propylene glycol, and ethylene glycol), 27% are n-alkanes, and the rest are largely branched and cyclic alkanes. The prominence of oxygenated IVOC emissions (e.g. siloxanes, benzyl alcohol, glycol ethers) from VCPs is noteworthy, as SOA yields from these compounds have not historically been evaluated nor included as SOA precursors in model chemical mechanisms (Qin et al., 2020). However, work has been undertaken in recent years to better understand these compounds (e.g. Wu and Johnson 2017; Li and Cocker 2018; Janechek et al., 2019; Charan et al., 2020). Overall, Paints &

Coatings is the largest source of IVOC emissions (~920 g person$^{-1}$ year$^{-1}$; Fig. 3), followed by Printing Inks (~350 g person$^{-1}$ year$^{-1}$), Cleaning Products (~180 g person$^{-1}$ year$^{-1}$), and Pesticides (~170 g person$^{-1}$ year$^{-1}$). While Paints & Coatings emit more IVOCs by mass than all other PUCs, Printing Ink and Pesticide emissions both feature greater proportions of IVOCs to their total emissions (~44% and ~29%, respectively).

These results also highlight how emissions from each PUC and sub-PUC are uniquely driven by mass of products used, organic composition, and use timescale. For example, the two largest sub-PUC sources are Daily Use Products and General Cleaners. Both are assigned a use timescale of 24-hr, but 40.6% of Daily Use Products are organic while General Cleaners are overwhelming composed of water (Table 2) and the annual mass usage of General Cleaners is ~3x higher than Daily Use





Products (Table 1). As a result, net emissions of General Cleaners are within 10% of those from Daily Use Products (1.85 kg
person$^{-1}$ year$^{-1}$ and 2.03 kg person$^{-1}$ year$^{-1}$, respectively). The emissions of Short Use Products, which is assigned a "Minutes"
use timescale, can further illustrate the importance of considering fate-and-transport. Under these use timescale assumptions,
only high volatility compounds (i.e. $C^* > 3 \times 10^7 \, \mu g/m^3$) are emitted and a majority (~97%) of its organics are retained
(Table 3). Besides Daily Use Products and General Cleaners, all remaining sub-PUCs emit $\leq$ 1.14 kg person$^{-1}$ year$^{-1}$, with six
emitting less than 0.1 kg person$^{-1}$ year$^{-1}$ (Table 3). Generally, sub-PUCs with low emissions stem from minimal use (e.g. Misc.
Products), short use timescales (e.g. Short Use Products), or high control assumptions (e.g. Oil & Gas, Fuels & Lighter).

**3.2 Uncertainty Analysis of National-Level Emission Factors**

Uncertainty associated with product usage, proportion of evaporative organics, assumptions related to evaporation and use
timescale, and post-use controls, where applicable, result in a total sector-wide emission uncertainty of ±15% (Fig. 4; 9.7 kg
person$^{-1}$ year$^{-1}$ [95% CI: 8.3 – 11.2]). Interestingly, the interaction of evaporation and use timescales can result in a threshold
effect, where small changes in either do not necessarily translate into changes in the magnitude of emissions for a given sub-PUC
(Fig. S2). For many PUCs, such as Paints & Coatings, Adhesives & Sealants, and Printing Inks, the use timescale is sufficiently
long (i.e. years) for all evaporative organics to evaporate, regardless of the uncertainty associated with the evaporation and use
timescales. Under such conditions, only uncertainty in product usage and product composition affect uncertainty in the emission
magnitude. As a result, these two variables are the largest drivers of uncertainty for the complete sector (Fig. S2). However,
uncertainties associated with evaporation and use timescale assumptions can be important for certain sub-PUCs with moderate to
low use timescales (see Cleaning Products in Fig. S2). For example, Detergents & Soaps is assigned a "Minutes" use timescale,
which results in a 0.12 kg person$^{-1}$ year$^{-1}$ emission factor (Table 3). If the use timescale for this sub-PUC was changed to 1-hr,
the emission factor would increase by a factor of 5.

From a national emissions perspective, these Monte Carlo results contain several important results. First, as mentioned above,
the largest drivers of uncertainty are associated with a sub-PUC's usage and composition, not assumptions related to fate-and-
transport (i.e. evaporation and use timescales). Second, the most uncertain PUCs are Cleaning Products, Personal Care Products,
and Paints & Coatings, and their uncertainty generates a significant amount of emissions potential. The 95% confidence interval
for all three span > 1.3 kg person$^{-1}$ year$^{-1}$, which is equivalent to > 400 Gg of organic emissions per year. Finally, the 95%
confidence interval for the national level emissions from the complete sector for 2016 is 2.7 – 3.6 Tg (1.8 – 2.4 TgC), which is
broadly consistent with the US EPA's 2017 NEI and, largely due to differences in predicted evaporation, approximately half the
emissions magnitude reported elsewhere (McDonald et al., 2018).

**3.3 State and County-Level Emissions Allocation**

The magnitude of VCP emissions varies substantially throughout the country, with the most populated states and counties
featuring the highest ROC emissions (Fig. 5). California (358 Gg), Texas (253 Gg), and Florida (177 Gg) are the largest state-
level emitters and contribute ~25% of all VCP emissions. In contrast, the 30 smallest state-level emitters (plus Washington, DC)
together emit ~800 Gg. At the county-level, Los Angeles County, Cook County (Chicago), and Harris County (Houston) are the
largest emitters. However, after normalizing by population, these three counties all feature per-capita emissions (8.42, 9.09, and
8.97 kg person$^{-1}$ year$^{-1}$, respectively) less than the national average (9.67 kg person$^{-1}$ year$^{-1}$) due to less industrial activity.


National spatial variability in per-capita emissions are largely driven by sub-PUCs tied to industrial and commercial activity





(Fig. 5). These sub-PUCs include Allied Paint Products (1.14 kg person$^{-1}$ year$^{-1}$), Industrial Coatings (1.04 kg person$^{-1}$ year$^{-1}$), Printing Inks (0.80 kg person$^{-1}$ year$^{-1}$), Agricultural Pesticides (0.53 kg person$^{-1}$ year$^{-1}$), and Oil & Gas (0.08 kg person$^{-1}$ year$^{-1}$). The employment proxies for Allied Paint Products, Industrial Coatings, and Printing Inks are usually consistent with the underlying population (Fig. S3), with peaks in California, Texas, Florida, New York, and the industrial Midwest. In contrast, emissions from Agricultural Pesticides and Oil & Gas drive the large per-capita emissions in the Midwest and Great Plains (Fig. 5). Emissions from these two sub-PUCs are heavily concentrated in the central United States (Fig. S3), including North Dakota, South Dakota, Iowa, Nebraska, Kansas, and Oklahoma. Collectively, these states contain < 4.5 % of the United States population but 24.1% and 17.5% of the Agricultural Pesticides and Oil & Gas VCP emissions, respectively. Both sub-PUCs also contribute to atypically high per-capita emissions in other States, such as Texas, Colorado, Idaho, and Wyoming.

While national VCP emissions from the 2017 NEI and the VCPy inventory are broadly consistent, county-level and categorical estimates can differ substantially between the two (Fig. S4). For example, 5% of all counties feature a decrease of > 35% and another 5% feature an increase of > 60%. When compared to the 2017 NEI, the states with the largest emissions increases were Delaware, California, and Colorado, and the States with the largest emissions decreases were North Dakota and South Dakota. There are also many spatial similarities between the two inventories. Both feature peaks in per-capita emissions over the Midwest and Great Plains (Fig. S4) and approximately half of all County-level emissions in the VCPy inventory are within 14% of their value in the 2017 NEI. To compare the two inventories categorically, all product use categories are mapped to individual Source Classification Codes (SCCs; Table S10). Categorically, Personal Care Products (150%) and Paints & Coatings (34%) feature the largest increases, whereas Pesticides (-54%) and Printing Inks (-13%) feature the largest decreases. The VCPy inventory also includes marginal increases in Cleaning Products and Adhesives & Sealants emissions, while also quantifying solvent-borne emissions in Oil & Gas operations (included as "Other" in Fig. S5).

### 3.4 Evaluation of Inventory Using Emission Ratios

Predicted per-capita VCP emissions in Los Angeles County are 8.42 kg person$^{-1}$ year$^{-1}$ and consist of 250+ organic compounds. Observed emission ratios were available for 30 species (Table S7), including some of the most abundantly emitted (e.g. ethanol, acetone, isopropyl alcohol, toluene). In fact, of the 30 available emission ratios, 24 were for compounds that contributed more than 0.1% to total VCP emissions (Fig. 6), providing the opportunity to evaluate important markers. For most compounds, the VCPy estimate was well within a factor of 2 when compared to observations. Some important markers were marginally low biased (e.g. ethanol, isopropyl alcohol), while others were marginally high biased (e.g. acetone, methyl ethyl ketone, isobutane), illustrating the difficulty in precisely speciating organic emissions and uncertainties introduced by compositing. However, when considered as a whole, the complete VCPy inventory performs remarkably well with a correlation of 0.95. In total, the observed emission ratio for all 30 compounds was 0.259 g (g CO)$^{-1}$ and the inventory estimate is 0.226 g (g CO)$^{-1}$, indicating a 13% low bias. In addition, the VCPy inventory shows a marked improvement over the 2017 NEI, which reports 3.28 kg person$^{-1}$ year$^{-1}$ of VCP emissions in Los Angeles County. For the 30 compounds considered here, the 2017 NEI reports 0.143 g (g CO)$^{-1}$, which is 45% lower than observations (Fig. S6). Most notably, the emissions ratio of ethanol, acetone, isopropyl alcohol, and propane, all of which are emitted by VCPs in substantial quantities, were low by a factor of 2-3.

While the residual, 13% low bias could suggest that additional organic emissions might be missing from the VCPy inventory, several other factors could explain discrepancies. First, emission ratios are equally sensitive to both organic and CO emissions. While CO appears to be represented and modelled well in current inventories (Lu et al., 2020), a marginal, systematic bias in CO





can affect the results presented here. For example, if the CO inventory were systematically high bias by 10%, the bias in the VCPy inventory emission ratios would be nearly eliminated. Second, since emission ratios are not sector-specific but reflect total emissions, missing organic emissions might be from other sources. Mobile sources, especially gasoline exhaust, is rich in small ($\leq C_6$) hydrocarbons, including ethene, n-butane, n-pentane, isopentane, methylpentanes, propene, and methylhexanes (Gentner

et al., 2013). Except for n-butane, none of these compounds appreciably come from VCP sources and all are low biased in the complete inventory (Fig. S6). Finally, while the ambient NMVOC to CO concentration ratio in Los Angeles has been consistent for several decades (McDonald et al., 2013), it is possible that trends for these two pollutants could have diverged in recent years.

### 3.5 Effective SOA Yields, O3 MIR, and Air Pollution Potential

Nationally, the effective SOA yield of the complete sector is 5.3% by mass (Table 4) and the most abundantly emitted SOA precursors are IVOC alkanes, aromatics, volatile methyl siloxanes, and fragrances. On a sub-PUC basis, the effective yield spans more than two-orders of magnitude, with Short Use Products and Printing Inks featuring an effective yield of 0.05% and 14.8%, respectively. For $O_3$, the effective MIR of the complete sector is 1.6 (g $O_3$) g$^{-1}$ and, when compared to SOA yields, there is considerably less sub-PUC variability. While VCPs do emit aromatics and alkenes, both of which are photochemically reactive

compound classes with high ozone potential, emissions are usually dominated by oxygenated compounds and alkanes, such as acetone, isopropyl alcohol, propane, and isobutane, which are minimally reactive. In fact, of the top fifteen highest emitting VCP compounds, seven feature a MIR < 1.0 (g $O_3$) g$^{-1}$.

While a sub-PUC may be a large source of organic emissions, this does not necessarily translate to a high potential impact on

PM$_{2.5}$ and ozone. This is best highlighted by Industrial and Architectural Coatings. Together, these two sub-PUCs constitute ~20% of all VCP emissions (Table 3), but only ~10% of the total SOA potential due to their low effective yields (2.94% and 2.42%, respectively). Architectural Coatings emissions feature significant quantities of Texanol™ (a highly branched oxygenate) and small glycols, such as propylene and ethylene glycol. A < 1% and 0% SOA yield is assigned to Texanol™ and both glycols, respectively. Though, it should be noted that this may be a lower bound as Li et al., 2018 report moderate aerosol formation from

propylene glycol. Similarly, Printing Inks contribute ~8% of all VCP emissions, which is nearly 2.5x less than Daily Use Products and General Cleaners nationally (Table 3). However, Printing Ink emissions are dominated by IVOC alkanes and aromatics, resulting in a high effective SOA yield (14.4%). As a result, Printing Inks contribute significantly to the total SOA potential nationally (Fig. 7). Paints & Coatings are nonetheless the dominant contributor to SOA potential, but this is more so due to the high emissions of the component sub-PUCs rather than their modest effective SOA yields (2.42 – 6.56%). Both

General Cleaners and Daily Use Products also have moderate quantities of SOA precursors and high emissions, which translates to 17.2% and 13.1% of the national VCP SOA potential, respectively. Since the effective MIR of each sub-PUC is not highly variable, $O_3$ potential is highly correlated with emissions magnitude. Overall, the three highest emitting PUC, Paints & Coatings, Cleaning Products, and Personal Care Products, are also the highest contributors to $O_3$ potential (Fig. 7).

These results also demonstrate how fate-and-transport assumptions can impact estimates of SOA production. For example, a prior study reported that both laundry detergent and a general-purpose spray cleaner can form appreciable quantities of SOA (Li et al., 2018). Here, the VCPy inventory reports an effective yield of 0.0% by mass of organic emitted for Detergents & Soaps and 4.7% for General Cleaners (Table 4). While the organic content of both sub-PUCs, by mass, is $\geq$ 18% (Table 2), Detergents & Soaps feature a dramatically smaller use timescale (Minutes vs. Days). As a result, not only is the total mass of organic emissions





from Detergents & Soaps smaller than General Cleaners, but the collection of compounds that are emitted feature systematically smaller evaporation timescales. Such compounds are highly volatile (i.e. $C^* > 1 \times 10^8 \, \mu g \, m^{-3}$) and not SOA precursors. In contrast, General Cleaners are assigned a longer use timescale, which provides time for lower volatility organics (i.e. IVOCs) to evaporate and subsequently contribute to the formation of SOA.

### 3.6 Non-Evaporative Organic Assumptions

The composition and volatility distribution of the organics assumed to be non-evaporative, which is ~60% of all organics (Fig. S8), is unidentified and assumed to be entirely non-volatile for the main analysis. However, there is evidence that a non-negligible portion of this mass may be SVOCs ($0.3 \, \mu g \, m^{-3} < C^* < 300 \, \mu g \, m^{-3}$), which can evaporate on atmospherically relevant timescales (Khare and Gentner, 2018). SHEDS-HT, a near-field model used to prioritize human exposure to chemicals (Isaacs et al., 2014), reports that > 15%, > 5%, and > 2% of all organics found in residential personal care product, household

product, and coatings, respectively, are composed of SVOCs (Qin et al., 2020). The treatment of non-evaporative organics and their potential emission can have a substantial impact on the modulation of SOA potential from VCPs. For example, if the assumption regarding evaporation of these organics is relaxed by assuming 1% of all non-evaporative organics eventually do evaporate, sector-wide emissions would increase by 0.18 kg person$^{-1}$ year$^{-1}$ (i.e. < 2% of the VCP emissions). Such a scenario is possible for products featuring long use timescales (e.g. paints, pesticides), if SVOCs are considered non-evaporative, or if

products featuring shorter use timescales (e.g. Daily Use Products, Cleaning Products) are not fully sequestered. Since this increase in emissions is minor (i.e. < 2%), there would be negligible impacts on the total emission magnitude and O$_3$. However, these compounds, by definition, feature low vapor pressures, which makes them prime SOA precursor candidates. If these compounds were permitted to form SOA with 100% efficiency, the effective yield from the complete sector would increase from 5.3 to 7.0% by mass (Fig. S8). Correspondingly, if 2% of all non-evaporative organics were assumed to evaporate with similar

SOA formation assumptions, the effective yield from the complete sector would increase to 8.6% by mass.

### 4 Additional Uncertainties

    The current VCPy framework assumes all evaporated organics reach the ambient atmosphere, regardless of origin. However, VCP emissions occur both indoors and outdoors (Farmer et al., 2019; Nazaroff and Weschler, 2004; Singer et al., 2006a). In fact, the indoor concentration of prevalent VCP markers and secondary pollutants often exceeds outdoor concentrations (Farmer et al.,

2019; Patel et al., 2020). For ambient air emissions, consideration of VCP emissions indoors is important if there is a gas-phase loss mechanism occurring at a scale that is comparable to typical indoor air exchange rates (~0.5 hr$^{-1}$; Murray and Burmaster, 1995). Indeed, sorption of gas-phase organics (e.g. terpenes) into typical residential furnishing and dust has been shown to occur on relevant timescales (Singer et al., 2007; Singer et al., 2004; Weschler and Nazaroff, 2008). Organics emitted indoors can also react with oxidants, leading to the formation of lower-volatility organics that can form particulates (Nazaroff and Weschler,

2004; Singer et al., 2006b). These particulates can deposit before outdoor exhaust can occur due to the high surface-to-volume ratio of indoor settings (Abbatt and Wang, 2020; Farmer et al., 2019). Planned future VCPy functionality includes the incorporation of a two-box model to capture these possible termination mechanisms and distinguish between near-field and far-field exposure pathways.

In addition, the efficiency of post-use controls for several sub-PUCs can be highly uncertain and vary both in space and time. In particular, this includes Oil & Gas, which is assigned a post-use control based on average reported reinjection rates of produced





water (Liden et al., 2018; Lyman et al., 2018), as well as Industrial Coatings and Printing Inks, which occur at facilities capable of add-on controls (U.S. Environmental Protection Agency, 2006a; 2006b; 2007; 2008). Here, post-use controls are not assigned for Industrial Coatings or Printing Inks. As such, emissions from these sub-PUCs could feature localized high bias, depending on

regional control requirements for facilities that use associated products. Similarly, the spatial allocation of nonpoint emissions features unique difficulties. For example, even if the allocation of nonpoint emissions was precisely matched to a quantifiable proxy, variation in the emission strength of individuals within that proxy (e.g. humans or employees) is often neglected (Li et al., 2020).

**5 Conclusions**

VCPy is a new framework to model organic emissions from volatile chemical products throughout the United States including spatial allocation to regional and local scales. In VCPy, product volatilization is a function of the characteristic evaporation timescale of individual components and the use timescale for product-use categories. National, per-capita organic emissions from VCPs are 9.7 kg person$^{-1}$ year$^{-1}$ (6.5 kgC person$^{-1}$ year$^{-1}$) for 2016, which translates to 3.12 Tg (2.10 TgC) for the U.S. Paints & Coatings, Personal Care Products, and Cleaning Products contribute most to these emissions. When filtered to remove regulatory

exempt organics, which enables a direct comparison to the EPA's NEI, total emissions from VCPs are 2.6 Tg of VOC and equal in magnitude to the sum of all mobile sources nationally, thus highlighting the growing importance of the VCP sector. Organic emissions featured substantial (~20%) contributions from IVOCs, which are likely SOA precursors. Of this 20%, 55% are oxygenated compounds, 27% are n-alkanes, and the rest are largely branched and cyclic alkanes. Nationally, the effective SOA yield and MIR, two metrics that facilitate an approximation of the potential air quality impacts, of VCPs is 5.3% by mass and

1.59 (g O$_3$) g$^{-1}$, respectively. This effective SOA yield indicates VCPs are likely a significant source of SOA in urban environments (Qin et al., 2020).

Uncertainty associated with this framework was tested through Monte Carlo analysis. Notably, the dominant drivers of uncertainty were associated with estimated product usage and the composition of products, and not assumptions related to fate-

and-transport. SOA formation from VCP emissions is especially sensitive to assumptions regarding evaporation of low volatility species. If 1% of all non-evaporative organics eventually do evaporate, sector-wide emissions would increase by 0.18 kg person$^{-1}$ year$^{-1}$ and the effective SOA yield from the complete sector could increase by > 1%. The 95% confidence interval for the national level emissions from the complete sector for 2016 is 2.68 – 3.60 Tg (1.81 – 2.42 TgC). This is consistent with the 2017 National Emission Inventory and half the emissions magnitude reported elsewhere (McDonald et al., 2018).


While the national level emissions from the VCPy framework and the 2017 NEI are comparable, regional and localized differences can be significant. This is most clear when evaluating the VCPy inventory to published emission ratios. For Los Angeles County, the VCPy inventory performs well (normalized mean bias of -13% with r = 0.95) and is significantly improved over the reported 2017 NEI VCP emissions. Planned future work includes adoption of variable emission settings (indoor vs.

outdoor) to account for loss mechanisms indoors (e.g. gas-phase sorption to surfaces), revisited mapping of VCP emissions to common chemical mechanisms for ease of research use in the chemical transport modelling community, estimation of SOA and ozone formation from VCPs using a chemical transport model and VCPy emissions inputs, and understanding the evolution of VCP emissions over time.



**Data Availability**

VCPy.v1.0 will be available on data.gov following publication (doi to be provided). All data presented in this manuscript can be retrieved and/or generated by downloading VCPy.v1.0. Additional instructions can be found in the main directory and guidance can be requested by contacting one of the corresponding authors.

**Author Contributions**

KMS and HOTP designed the research scope. All authors participated in data curation and/or analysis. KMS and HOTP drafted
the initial manuscript and all authors contributed to subsequent drafts.

**Competing Interests**

The authors declare that they have no conflicts of interest.

**Disclaimer**

Although this work was contributed by research staff in the Environmental Protection Agency and has been reviewed and
approved for publication, it does not reflect official policy of the EPA. The views expressed in this document are solely those of authors and do not necessarily reflect those of the Agency. EPA does not endorse any products or commercial services mentioned in this publication.

**Acknowledgements**

The authors would like to thank Janice Godfrey, Art Diem, Jennifer Snyder, Rich Mason, Caroline Farkas, Claudia Toro, Alison
Eyth, Luke Valin, Mohammed Jaoui, Jim Szykman, Donna Schwede, Christian Hogrefe, Kristen Foley, Jesse Bash, Marc Houyoux, and Cindy Beeler at the U.S. EPA and Kyriacos Kyriacou and Jose Gomez at the California Air Resources Board for helpful discussions and/or data acquisition. Comments by Marc Houyoux (EPA), Jim Szykman (EPA), and ## anonymous reviewers served to strengthen this manuscript.

**Financial Support**

Karl Seltzer and Elyse Pennington were supported by the Oak Ridge Institute for Science and Education (ORISE) Research Participation Program for the U.S. Environmental Protection Agency (EPA).

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



**Table 1: Description of all PUCs and sub-PUCs currently implemented in VCPy, their estimated mass usage for 2016, and product examples of each. See Table S2 for a derivation of all product usage estimates.**

| Product Use Categories (PUCs) | Sub-Product Use Categories (sub-PUCs) | 2016 Annual Usage [kg person$^{-1}$ year$^{-1}$] | Product Examples |
|---|---|---|---|
| Cleaning Products | Detergents & Soaps | 40.58 | Soaps, Detergents, Metal Cleaners, Scouring Cleaners |
| | General Cleaners | 28.47 | Disinfectants, Air Fresheners, Glass & Bathroom Cleaners, Windshield Washer Fluid, Hand Sanitizer, Automotive & Floor Polishes, Bleaches, Surfactants |
| Personal Care Products | Daily Use Products | 8.83 | Hair Products, Perfumes, Colognes, Cleansing & Moisturizing Creams, Sunscreens, Hand & Body Lotion and Oils, Cosmetics, Deodorants |
| | Short Use Products | 3.16 | Shampoo, Conditioners, Shaving Cream, Aftershave, Mouthwashes, Toothpaste |
| Adhesives & Sealants | Adhesives & Sealants | 15.23 | Glues and Adhesives, Epoxy Adhesives, Other Adhesives, Structural and Nonstructural Caulking Compounds and Sealants |
| Paints & Coatings | Architectural Coatings | 13.27 | Exterior/Interior Flat/Gloss Paints, Primers, Sealers, Lacquers |
| | Aerosol Coatings | 0.39 | Paint Concentrates Produced for Aerosol Containers |
| | Allied Paint Products | 1.26 | Thinners, Strippers, Cleaners, Paint/Varnish Removers |
| | Industrial Coatings | 7.42 | Automotive, Appliance, Furniture, Paper, Electrical Insulating, Marine, Maintenance, and Traffic Marking Finishes and Paints |
| Printing Inks | Printing Inks | 3.20 | Letterpress, Lithographic, Gravure, Flexographic, Nonimpact/Digital Inks |
| Pesticides & FIFRA Products | FIFRA Pesticides | 1.46 | Lawn and Garden Pesticides and Chemicals, Household and Institutional Pesticides and Chemicals |
| | Agricultural Pesticides | 10.32 | Agricultural and Commercial Pesticides & Other Organic Chemicals |
| Dry Cleaning | Dry Cleaning | 0.03 | Dry Cleaning Fluids |
| Oil & Gas | Oil & Gas | 1.32 | Cleaners, Deicers |
| Misc. Products | Misc. Products | 0.18 | Pens, Markers, Arts and Crafts, Dyes |
| Fuels & Lighter | Fuels & Lighter | 2.80 | Lighter Fluid, Fire Starter, Other Fuels |





**Table 2: 1ˢᵗ-Order product composition profiles and evaporative organics proportion for all sub-PUCs.**

| Product Use Categories (PUCs) | Sub-Product Use Categories (sub-PUCs) | Water | Inorganic | Non-Evaporative Organics[a] | Evaporative Organics[a] |
|---|---|---|---|---|---|
| Cleaning Products | Detergents & Soaps[b] | 67.8% | 13.9% | 15.4% | 2.9% |
| | General Cleaners[b] | 73.3% | 8.6% | 11.1% | 6.9% |
| Personal Care Products | Daily Use Products[b] | 48.8% | 10.7% | 16.9% | 23.7% |
| | Short Use Products[b] | 72.2% | 5.8% | 17.7% | 4.3% |
| Adhesives & Sealants | Adhesives & Sealants[b] | 12.8% | 53.2% | 29.0% | 5.0% |
| Paints & Coatings | Architectural Coatings[c] | 41.9% | 51.1% | 0.0% | 6.8% |
| | Aerosol Coatings[d] | 12.7% | 12.7% | 0.0% | 74.7% |
| | Allied Paint Products[b] | 5.1% | 3.5% | 0.6% | 90.8% |
| | Industrial Coatings[e] | 15.0% | 70.0% | 0.0% | 14.0% |
| Printing Inks | Printing Inks[f] | 8.0% | 67.0% | 0.0% | 25.0% |
| Pesticides & FIFRA Products | FIFRA Pesticides[b] | 74.8% | 4.9% | 15.1% | 5.1% |
| | Agricultural Pesticides[b] | 74.8% | 4.9% | 15.1% | 5.1% |
| Dry Cleaning | Dry Cleaning[g] | 0.0% | 0.0% | 0.0% | 100% |
| Oil & Gas | Oil & Gas[g] | 0.0% | 0.0% | 0.0% | 100% |
| Misc. Products | Misc. Products[b] | 27.1% | 14.6% | 48.8% | 9.5% |
| Fuels & Lighter | Fuels & Lighter[b] | 0.0% | 92.9% | 0.0% | 7.1% |

[a]: "Non-Evaporative Organics" and "Evaporative Organics" sum to total product organics. "Evaporative Organics" represent the potentially evaporative organic fraction of the total product and excludes assumed "non-evaporative" (i.e. assumed non-volatile) organics, which are not included in the California Air Resource Board's organic profiles.

[b]: Source: California Air Resources Board 2015 Consumer and Commercial Products Survey Data (CARB, 2019).

[c]: Source: California Air Resources Board 2005 Architectural Coatings Survey (CARB, 2007). VOC + Exempts is used for both organic and evaporative organics. Non-evaporative organic proportions not provided.

[d]: Source: California Air Resources Board 2010 Aerosol Coatings Survey (CARB, 2012). Only evaporative organics is provided. Remainder (~25%) is split evenly between water and inorganics.

[e]: Source: Industrial Maintenance composition data from California Air Resources Board 2005 Architectural Coatings Survey (CARB, 885   2007).

[f]: Source: Graphic Arts composition data from California Air Resources Board 2005 Architectural Coatings Survey (CARB, 2007).

[g]: All product usage is composed of organic functional solvents (The Freedonia Group, 2016). Therefore, all mass is assumed to be potentially evaporative.


**Table 3: National-level emissions, volatilization fraction, and proportion of all usage that is emitted for all sub-PUCs.**

| Product Use Categories (PUCs) | Sub-Product Use Categories (sub-PUCs) | ROC Emissions | | Organic Volatilization Fraction [%][a] | Total Product Emitted [%] |
|---|---|---|---|---|---|
| | | [kg person⁻¹ year⁻¹] | [kgC person⁻¹ year⁻¹] | | |
| Cleaning Products | Detergents & Soaps | 0.12 | 0.06 | 1.6% | 0.3% |
| | General Cleaners | 1.85 | 1.25 | 36.0% | 6.5% |
| Personal Care Products | Daily Use Products | 2.03 | 1.12 | 56.7% | 23.0% |
| | Short Use Products | 0.02 | 0.01 | 3.3% | 0.7% |
| Adhesives & Sealants | Adhesives & Sealants | 0.76 | 0.56 | 14.7% | 5.0% |
| Paints & Coatings | Architectural Coatings | 0.89 | 0.51 | 100%[b] | 6.7% |
| | Aerosol Coatings | 0.29 | 0.22 | 100%[b] | 74.7% |
| | Allied Paint Products | 1.14 | 0.80 | 99.2% | 90.6% |
| | Industrial Coatings | 1.04 | 0.79 | 100%[b] | 14.0% |
| Printing Inks | Printing Inks | 0.80 | 0.65 | 100%[b] | 25.0% |
| Pesticides & FIFRA Products | FIFRA Pesticides | 0.07 | 0.06 | 25.2% | 5.1% |
| | Agricultural Pesticides | 0.53 | 0.41 | 25.2% | 5.1% |
| Dry Cleaning | Dry Cleaning | 0.01 | 0.01 | 34.5% | 34.5% |
| Oil & Gas | Oil & Gas | 0.08 | 0.04 | 6.0% | 6.0% |
| Misc. Products | Misc. Products | 0.02 | 0.01 | 16.3% | 9.5% |
| Fuels & Lighter | Fuels & Lighter | 0.02 | 0.02 | 10.0% | 0.7% |
| Total | | 9.67 | 6.52 | 32.0% | 7.0% |

[a]: Volatilization fraction represents the fraction of the total organic content of products that volatilize/emit to ambient air.

[b]: The "Organic" portion of these sub-PUCs is entirely composed of "Evaporative Organics" (see Table 2). Only data from the California Air Resources Board's 2015 Consumer and Commercial Products Survey featured the disaggregation of evaporative and non-evaporative organics. Prior surveys typically combined the non-evaporative organic portion of each profile with solids/inorganics.





**Table 4: The national effective SOA yield and MIR for all sub-PUCs. These results are plotted in Fig. S7.**

| Product Use Categories (PUCs) | Sub-Product Use Categories (sub-PUCs) | Effective SOA Yield [%] | Effective MIR [(g O$_3$) g$^{-1}$] |
|---|---|---|---|
| Cleaning Products | Detergents & Soaps | 0.00 | 1.48 |
| | General Cleaners | 4.74 | 1.88 |
| Personal Care Products | Daily Use Products | 3.27 | 1.38 |
| | Short Use Products | 0.05 | 1.27 |
| Adhesives & Sealants | Adhesives & Sealants | 6.19 | 1.51 |
| Paints & Coatings | Architectural Coatings | 2.42 | 1.89 |
| | Aerosol Coatings | 3.26 | 1.66 |
| | Allied Paint Products | 6.56 | 1.27 |
| | Industrial Coatings | 2.94 | 1.71 |
| Printing Inks | Printing Inks | 14.81 | 1.93 |
| Pesticides & FIFRA Products | FIFRA Pesticides | 8.10 | 1.01 |
| | Agricultural Pesticides | 8.10 | 1.01 |
| Dry Cleaning | Dry Cleaning | 3.47 | 1.13 |
| Oil & Gas | Oil & Gas | 2.21 | 1.03 |
| Misc. Products | Misc. Products | 1.94 | 2.26 |
| Fuels & Lighter | Fuels & Lighter | 5.35 | 1.15 |
| Total | | 5.26 | 1.59 |





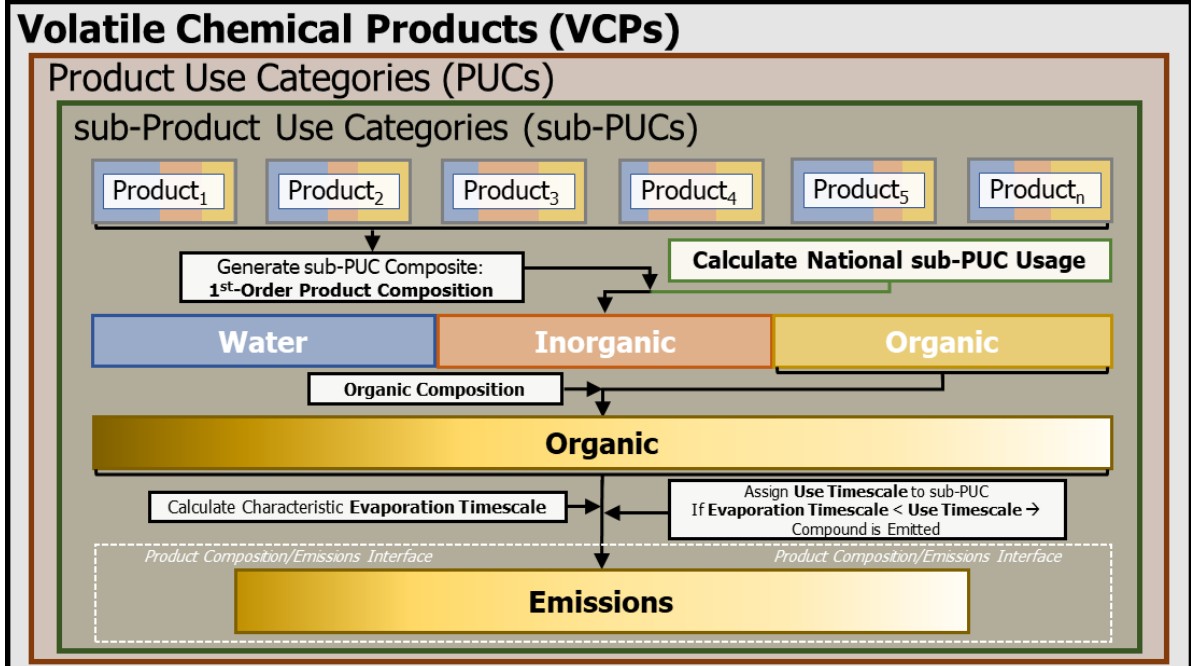

**Figure 1: Conceptual overview of the VCPy framework. Note: PUC = Product Use Category.**



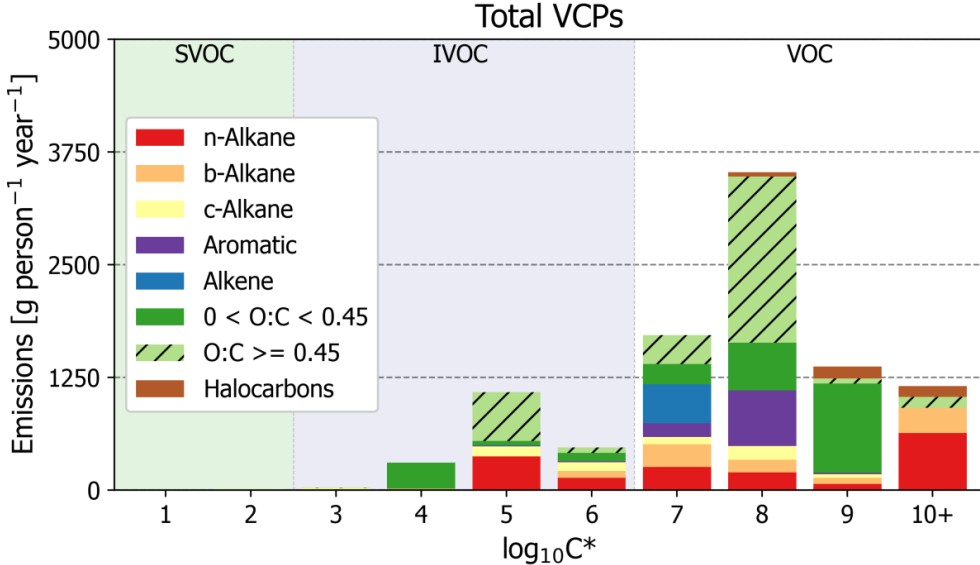


**Figure 2: Sector-wide volatility distribution of emissions by compound class.**




**Figure 3: PUC and sector-wide volatility distribution of organic emissions. Other is summation of Dry Cleaning, Oil & Gas, Misc. Products, and Fuels & Lighter. Pie charts are 1st-order product composition and organic emission proportions for PUCs and the complete sector. Note: The "Organic" portion of all Paints & Coatings and Printing Inks pie charts is entirely composed of "Evaporative Organics" (see Table 2).**





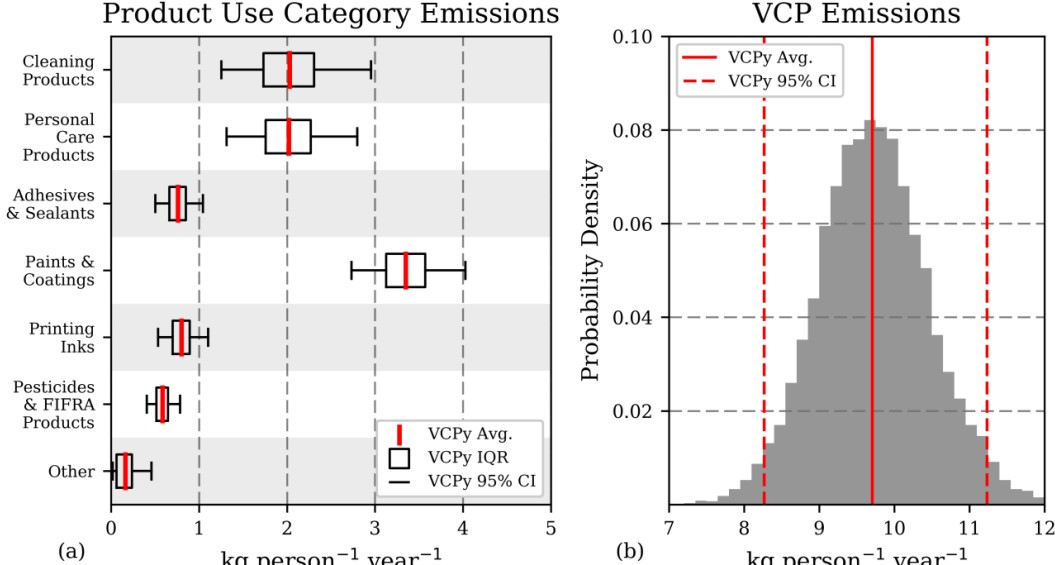

**Figure 4: Monte Carlo sensitivity results for organic emissions. (a) Mean, interquartile range, and 95% confidence intervals for six PUCs and a combination of the remaining four (Dry Cleaning, Oil & Gas, Misc. Products, and Fuels & Lighter). (b) Probability distribution of sector-wide emission estimates. See Table S9 for a tabulation of this figure.**



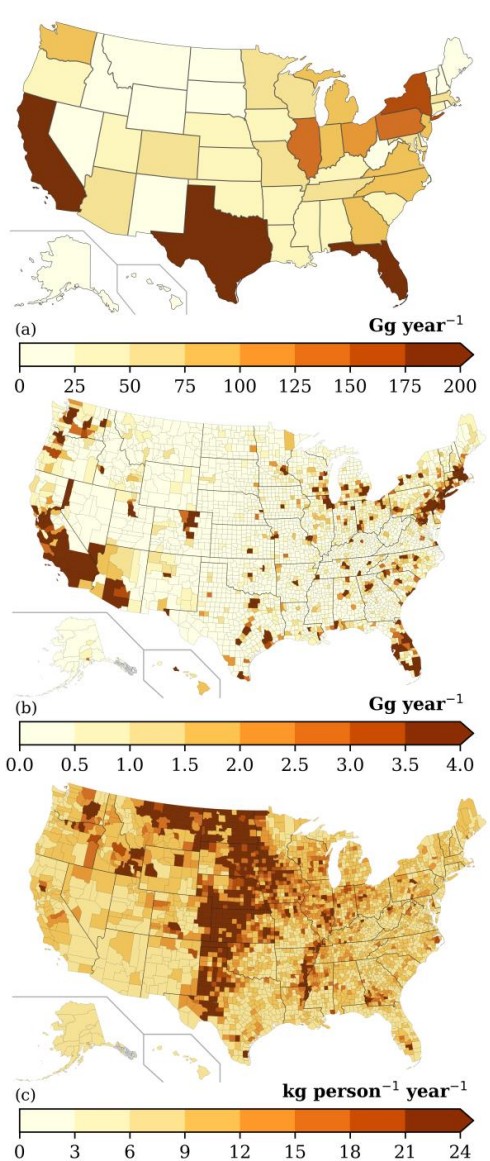

**Figure 5: (a) State-level, (b) County-level, and (c) County-level per-capita VCP emissions.**





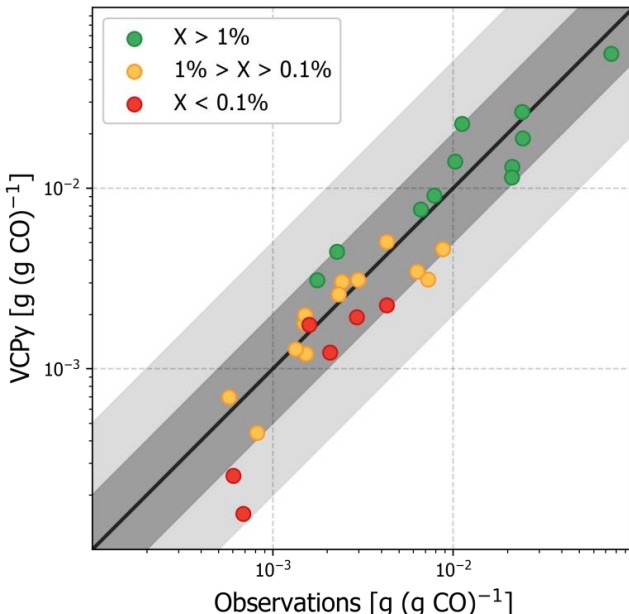

**Figure 6: Evaluation of organic emission ratios in Los Angeles County using observed emission ratios from summer 2010. VCPy inventory ratios utilize VCPy predicted emissions for VCPs and the 2017 NEI for all other sources. The scatter point colors represent the relative abundance of each compound in the complete VCP sector. For example, all green points represent compounds that are > 1% of the total VCP emissions in Los Angeles County. Black line – 1:1; Dark grey shading – 2:1; Light grey shading – 5:1. Values available in Table S7.**





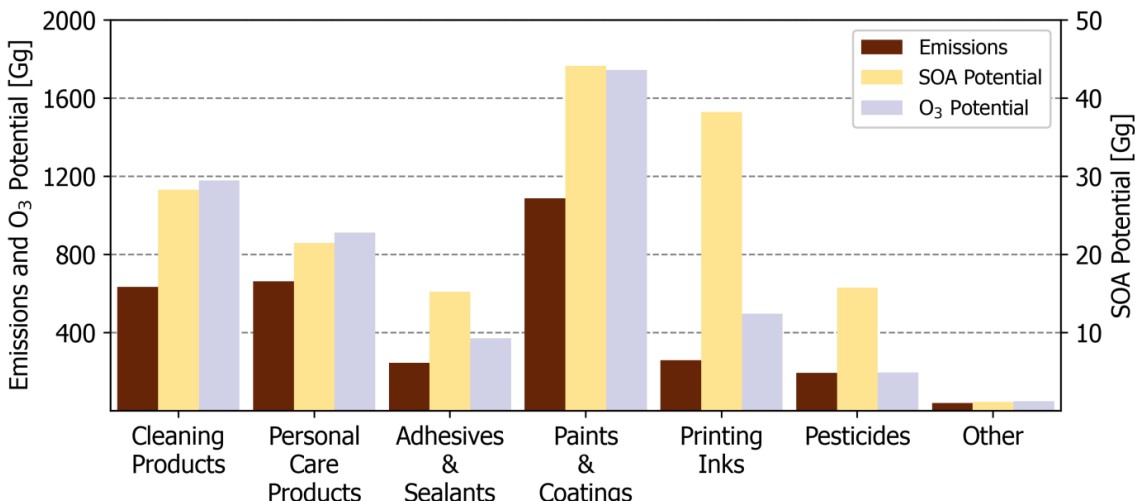

**Figure 7: National-level emissions, SOA potential, and O₃ potential by PUC. Other is summation of Dry Cleaning, Oil & Gas, Misc. Products, and Fuels & Lighter.**