# Peer review of "Reactive Organic Carbon Emissions from Volatile Chemical Products"

_Atmospheric Chemistry and Physics, 2020_

## Referee Comment (RC1) · Anonymous Referee #1 · 7 Dec 2020

The authors describe a new framework to model organic emissions from VCPs; including spatial allocation. This approach is novel in the fact that product volatilization is a function of the characteristic evaporation timescale of individual components physiochemical properties. National VCP emissions for 2016 were estimated to be 2.68 – 3.60 Tg (1.81 – 2.42 TgC) which was comparable to 2017 NEI values. The study highlights uncertainty from estimated product usage, product composition and indoor/outdoor settings. The article is well structured and clear. Given the importance of this sector and the need to resolve differences among various inventories it should be considered for publication with minor revisions.

Comments/Questions

1.) One of the main conclusions the authors make is that this new framework includes

spatial allocation to regional and local scales. Have you compared this to current surrogates provided with the 2017 NEI, CARB surrogates or published work such as "Improving spatial surrogates for area source emissions inventories in California" by Li et al. 2020? How do the regional and local distributions vary with this approach? What is the level of resolution the census data is applied to? County/census block? Possibly adding a difference plot comparing to current estimates would be helpful.

2.) Figure 5c shows a high amount of emissions per capita in Colusa, CA – what is the driver behind this in a relatively small county?

3.) Since observed data is available to do comparison, it would be beneficial to show a range of predicted VCP emissions for LA county of the 30 reported species. It is noted that the observed total is 0.259 g while the inventory total is 0.226g; can you add uncertainty to the inventory value based on the discussions from sector 3.6 and 4?

4.) In Section 5. on line 562 "The 95% confidence interval for the national level emissions from the complete sector for 2016 is 2.68 – 3.60 Tg (1.81 – 2.42 TgC). This is consistent with the 2017 National Emission Inventory and half the emissions magnitude reported elsewhere (McDonald et al., 2018)." Can the authors provide the 2017 NEI values that are being compared? It would also be helpful to add a national difference plot showing the variability between this new method and 2017 NEI totals for the three panels on figure 5 (state, county, county/capita).

---

## Referee Comment (RC2) · Anonymous Referee #2 · 29 Dec 2020

In this work, the authors developed a new framework for building emissions inventories of volatile chemical products (VCPs). VCPs have recently been recognized as a major, if not dominant, source of reactive organic carbon in urban areas. Since much of the focus of emission controls has historically been placed on mobile sources, accurate emission inventories are now needed for VCPs. In this work, the authors consolidated different tracking methods for a wide variety of product use categories under a unifying framework ("VCPy"), intended to be used by air quality models to predict ozone and secondary organic aerosol (SOA) formation. They also compared their emission prediction with field measurements and found good agreement when normalized to carbon monoxide, which gives credence to the method. Monte-Carlo analysis was also performed to explore uncertainties.

[Figure]

Overall, the manuscript was well written and I expect this framework will be used by many in the field of air quality. The description of the method is very detailed, which is necessary for this manuscript to provide good reference for future use. I recommend that this manuscript be published. I only have minor suggestions for the authors to consider, mostly for the purpose of discussion.

1. What is the role of disposal in this framework? If VCPs that are used on a short timescale but are disposed of using open methods, is that included in the framework? For example, VCPs could enter wastewater treatment plants and enter the atmosphere.

2. The Monte Carlo analysis is focused on the uncertainty in the total emissions per capita as the primary outcome. The assumption here is the uncertainty lies primarily in the model inputs, and the outcome is deterministic. I expect this assumption to be valid for total emission per capita as the primary outcome, but may not be so if we examine the composition instead. For example, how does uncertainty in the composition profile affect the emissions?

3. Similarly, for the input variables that were examined (e.g. uncertainty in v_e, depth), what is the uncertainty in the composition of emissions? E.g. what is the uncertainty in median c*? I expect that if v_e increases, it might increase emissions of lower volatility compounds more than it increases those of higher volatility ones.

4. For many of the water-based VCPs, I would expect that evaporation will be based more on K_AW (or Henry's Law constant) rather than K_OA. How much would that change the estimates?

5. What is the fraction of VCPs that are based on fossil-carbon vs modern carbon? Is that something that can be estimated?

6. Lines 296-297: For 4 PUCs, employment statistics were used for the spatial allocation of commercial VCP emissions. I am wondering that regarding the automation considerations, the number of employees might underestimate the VCP emissions from

those sites and sales allocation or GDP of the production sector distribution might be better tools for that purpose.

7. It is not surprising that regional and localized differences are significant. For many of the compounds, the atmospheric lifetimes could be long enough that these differences probably do not matter too much. This might also depend on the scale of air quality modeling.

Minor comments:

In Lines 25-27 (abstract) and in Section 3.3, when comparing to 2017 NEI, the terms "increase" and "decrease" are misleading, since they can be confused with year-to-year increase/decrease. I suggest using terms like "overestimate" or "underestimate", or just "higher" or "lower".

Line 80: there are two references for Li et al. 2018.

Figure 6: what does X stand for?

---

## Author Response (AR1)

**Response Letter to Reviewers for:**

**Reactive Organic Carbon Emissions from Volatile Chemical Products**

Karl M. Seltzer[1], Elyse Pennington[2,3], Venkatesh Rao[4], Benjamin N. Murphy[5], Madeleine Strum[4], Kristin K. Isaacs[5], Havala O.T. Pye[5]

[1]Oak Ridge Institute for Science and Education Postdoctoral Fellow in the Office of Research and Development, US Environmental Protection Agency, Research Triangle Park, NC 27711
[2]Oak Ridge Institute for Science and Education Fellow in the Office of Research and Development, US Environmental Protection Agency, Research Triangle Park, NC 27711
[3]California Institute of Technology, Pasadena, CA 91125
[4]Office of Air and Radiation, US Environmental Protection Agency, Research Triangle Park, NC 27711
[5]Office of Research and Development, US Environmental Protection Agency, Research Triangle Park, NC 27711

*Correspondence to*: Havala O.T. Pye (Pye.Havala@epa.gov)

Thank you to the editor and reviewers for taking the time to consider our manuscript and provide helpful comments. These comments significantly improved the rigor and quality of our manuscript, and your time and efforts are much appreciated. All reviewer comments are addressed individually below, point-by-point, with the original comments featured in bold text and our response followed in non-bold text.

All updates to the original submission were tracked in the revised submission.

Anonymous Referee #1:

**One of the main conclusions the authors make is that this new framework includes spatial allocation to regional and local scales. Have you compared this to current surrogates provided with the 2017 NEI, CARB surrogates or published work such as "Improving spatial surrogates for area source emissions inventories in California" by Li et al. 2020? How do the regional and local distributions vary with this approach? What is the level of resolution the census data is applied to? County/census block? Possibly adding a difference plot comparing to current estimates would be helpful.**

We used the same spatial allocation methods as the 2017 NEI. This largely includes population and employment-based allocation, as outlined in Table S6 of the SI. For employment-based allocation, we match the employment NAICS codes used by the 2017 NEI, as outlined in the Technical Source Documentation of the inventory.

In addition, we are familiar with Li et al. 2020 and cite the paper in our manuscript. However, there is a key difference in their analysis and our applications. Namely, application of their spatial surrogates is at the sub-County level for photochemical modeling purposes, whereas we use spatial proxies only to downscale national-level emissions to the county-level.

Comparing possible inventory differences due to variable spatial surrogates, especially at the sub-County level, was outside the scope of this analysis. Nonetheless, leveraging the results from Li et al. 2020 would certainly be considered for those types of applications.

**Figure 5c shows a high amount of emissions per capita in Colusa, CA – what is the driver behind this in a relatively small county?**

Colusa County is sparsely populated, but relative to their population, features high agricultural pesticide usage. As a result, while small in magnitude (see Colusa County in Fig. 5b), their per-capita emissions are

quite high.

**Since observed data is available to do comparison, it would be beneficial to show a range of predicted VCP emissions for LA county of the 30 reported species. It is noted that the observed total is 0.259 g while the inventory total is 0.226g; can you add uncertainty to the inventory value based on the discussions from sector 3.6 and 4?**

It would be difficult to confidently show a range of predicted individual compounds for the evaluation since the uncertainty associated with the evaporative organic composition of individual product types is not known or provided by the source data. Our uncertainty analysis using Monte Carlo simulations focused on the total emission magnitude and did not perturb the relative abundance of an individual species in a product composition. For example, we do not know the uncertainty associated with the assignment of x% of toluene within a product type. One sentence regarding this limitation has been added to Section 2.2 and are included below:

*Furthermore, the uncertainty associated with the evaporative organic composition of individual product types is not known or provided by the source data.*

Additional emissions that could result from the discussion in Section 3.6 would not yield a change to the inventory evaluation. Since the evaluation is exclusive to 30 compounds, and these compounds are not influenced by the volatility (or emission) of the organics that are assumed to be non-evaporative, the predicted $0.226 \text{ g (g CO)}^{-1}$ would not change. As for the uncertainties discussed in Section 4, these would be difficult to translate without (1) incorporation of a two-box model into the framework to account for indoor loss, which is planned for future work or (2) localized post-use control status, which was outside the scope of this analysis.

**In Section 5. on line 562 "The 95% confidence interval for the national level emissions from the complete sector for 2016 is 2.68 – 3.60 Tg (1.81 – 2.42 TgC). This is consistent with the 2017 National Emission Inventory and half the emissions magnitude reported elsewhere (McDonald et al., 2018)." Can the authors provide the 2017 NEI values that are being compared? It would also be helpful to add a national difference plot showing the variability between this new method and 2017 NEI totals for the three panels on figure 5 (state, county, county/capita).**

We have added 2017 NEI values (2.84 Tg) for comparative purposes to both Section 5 and Section 3.2. We also direct the referee to Figure S4c in the SI for a county-level per-capita difference plot between the VCPy inventory and the 2017 NEI.

**Anonymous Referee #2:**

**What is the role of disposal in this framework? If VCPs that are used on a short timescale but are disposed of using open methods, is that included in the framework? For example, VCPs could enter wastewater treatment plants and enter the atmosphere.**

In this framework, disposal is treated as a permanent sequestration of organics. Research does suggest that organics entering a wastewater treatment plant are largely removed through biodegradation or sorption to sludge (Shin et al. 2015). However, there is significant aeration at such plants, providing opportunity for air release. The NEI does report emissions from Publicly Owned Treatment Works, but given the massive quantity of organics that feature this down-the-drain fate (organics from shampoos, soaps, conditioners, detergents), even marginal changes in the emission of organics from VCPs through this route could have notable impacts.

**The Monte Carlo analysis is focused on the uncertainty in the total emissions per capita as the primary outcome. The assumption here is the uncertainty lies primarily in the model inputs, and the outcome is deterministic. I expect this assumption to be valid for total emission per capita as the primary outcome, but may not be so if we examine the composition instead. For example, how does uncertainty in the composition profile affect the emissions?**

We do partially include uncertainty in the composition of VCPs in our Monte Carlo assumptions. This is accomplished by assigning uncertainty in the evaporative organic proportions of each sub-PUC, as noted in Section 2.2. However, due to data limitations, we do not assign uncertainty to the composition of evaporative organics. Nonetheless, changes in the organic composition of the evaporative organics would result in changes in the volatility distribution of the organics, which is implicitly accounted for via assigned uncertainty to the characteristic evaporation timescale.

**Similarly, for the input variables that were examined (e.g. uncertainty in v_e, depth), what is the uncertainty in the composition of emissions? E.g. what is the uncertainty in median c*? I expect that if v_e increases, it might increase emissions of lower volatility compounds more than it increases those of higher volatility ones.**

This is a great question. With available data, the uncertainty associated with the composition of the emissions would be extremely difficult to determine. For example, we do not know the uncertainty associated with the assignment of x% of toluene within a product type. However, we can test the second question. Our median $\log(C^*)$ for the emissions from the complete sector is 7.6 and the 95% uncertainty associated with that number due to the model inputs we can perturb (including $v_e$) is 7.5 – 7.8. This is certainly an underestimate in the uncertainty since, again, we do not know how the uncertainty associated with the composition of individual product types.

As for $v_e$, if this variable were increased, the characteristic evaporation timescale of all components would increase, assuming all other variables are held constant. If this were to happen, all compounds that are already predicted to evaporate (i.e. higher volatility ones) would still evaporate. However, as the reviewer points out, this could have notable impacts on lower volatility compounds that were previously predicted to not evaporate.

**For many of the water-based VCPs, I would expect that evaporation will be based more on K_AW (or Henry's Law constant) rather than K_OA. How much would that change the estimates?**

When using the QSAR predicted $K_{OA}$ and $K_{OW}$ and equation 3.4 of Weschler and Nazaroff (2008) for two compounds (propylene glycol and PCBTF), we yield a calculated $K_{WA}$ that is larger for one (propylene glycol), indicating less partitioning to air, and one that is smaller (PCBTF), indicating more partitioning to air. However, the change in the subsequent characteristic evaporation timescale from such changes seems to be only relevant for VCPs that would feature small to medium use timescales (e.g. < 1 day). For VCPs with a long use timescale (e.g. architectural coatings), this change would not yield a different conclusion regarding the proportion of emissions.

**What is the fraction of VCPs that are based on fossil-carbon vs modern carbon? Is that something that can be estimated?**

This is difficult to determine. Some compounds emitted from VCPs (e.g. ethanol, many fragrances) are likely dominated by modern carbon, whereas others (e.g. mineral spirits) are likely dominated by fossil-carbon. However, it is our understanding that the demand for "green" solvents has and continues to grow. According

to an industry study (The Freedonia Group, 2016), approximately 15% of functional solvent usage is derived from natural or renewable resources.

**Lines 296-297: For 4 PUCs, employment statistics were used for the spatial allocation of commercial VCP emissions. I am wondering that regarding the automation considerations, the number of employees might underestimate the VCP emissions from those sites and sales allocation or GDP of the production sector distribution might be better tools for that purpose.**

Since emission more likely occurs at or near the time of use, we prefer employment (for industrial sub-PUCs) as a spatial downscaling proxy over production sector statistics. In addition, larger production facilities are more likely to have add-on controls. While we did not consider add-on controls in this manuscript, emissions from these sources are likely non-linear with production volume.

**It is not surprising that regional and localized differences are significant. For many of the compounds, the atmospheric lifetimes could be long enough that these differences probably do not matter too much. This might also depend on the scale of air quality modeling.**

We agree.

**In Lines 25-27 (abstract) and in Section 3.3, when comparing to 2017 NEI, the terms "increase" and "decrease" are misleading, since they can be confused with year-to year increase/decrease. I suggest using terms like "overestimate" or "underestimate", or just "higher" or "lower".**

We adjusted both the text in the abstract and Section 3.3 to help alleviate this confusion. The two relevant sentences in the abstract now reads:

*VCPy predicts more VCP emissions than the NEI for approximately half of all counties, with 5% of all counties featuring emissions more than 55% higher. Categorically, VCPy reports higher emissions for personal care products (150%) and paints/coatings (25%) when compared to the NEI, whereas pesticides (-54%) and printing inks (-13%) feature lower emissions.*

**Line 80: there are two references for Li et al. 2018.**

We believe this is a mistake on the part of the reviewer. We refer to the first reference as "Li et al., 2018" in the text and the second reference as "Li and Cocker 2018," since there are only two authors of that manuscript.

**Figure 6: what does X stand for?**

The "X" represents each compound in the figure. A clarification has been added to the Fig. 6 caption.

**Additional edits:**

Following submission, we learned of more recent sales proportions for water vs. solvent-based architectural coatings in a California Air Resource Board survey. As a result, we updated the water vs. solvent-based proportions from 88/12% to 94/6% (i.e. the proportion of water-based coatings to total architectural coatings increased). These changes had marginal impacts on the complete inventory (~2%) and modest impacts on the architectural coatings sub-PUC (~25%).

[revised manuscript text omitted]
|  | Aerosol Coatings | Product type composite Composite derived from CARB's 2010 Aerosol Coatings Survey[d]Survey[f] and speciated using CARB organic profiles[c] |
|  | Allied Paint Products | Product type composite derived from CARB's 2015 Consumer and Commercial Products Survey[b] and speciated using CARB organic profiles[c]Composite derived from CARB's 2015 Consumer and Commercial Products Survey |
|  | Industrial Coatings | SPECIATEv5.0[e-0g] Profile: 3149 |
| Printing Inks | Printing Inks | SPECIATEv5.0[g] Profile: 2570 |

[revised manuscript text omitted]

100 **Table S10: Tabulation of Fig. 4 from main text.**

[revised manuscript text omitted]

---

## Author Response (AR2)

**Response Letter to Editor for:**

**Reactive Organic Carbon Emissions from Volatile Chemical Products**

Karl M. Seltzer[1], Elyse Pennington[2,3], Venkatesh Rao[4], Benjamin N. Murphy[5], Madeleine Strum[4], Kristin K. Isaacs[5], Havala O.T. Pye[5]

[1]Oak Ridge Institute for Science and Education Postdoctoral Fellow in the Office of Research and Development, US Environmental Protection Agency, Research Triangle Park, NC 27711
[2]Oak Ridge Institute for Science and Education Fellow in the Office of Research and Development, US Environmental Protection Agency, Research Triangle Park, NC 27711
[3]California Institute of Technology, Pasadena, CA 91125
[4]Office of Air and Radiation, US Environmental Protection Agency, Research Triangle Park, NC 27711
[5]Office of Research and Development, US Environmental Protection Agency, Research Triangle Park, NC 27711

*Correspondence to*: Havala O.T. Pye (Pye.Havala@epa.gov)

Thank you to the editor for reviewing our manuscript and suggesting changes that provide additional clarity. Nearly all revisions were adopted, as suggested. Please see below where all notes are addressed, point-by-point.

**Editor:**

**lines 16-19 are confusing as written (and in the introduction). Part of the confusion is that it is unclear what "these processes" refer to and it is easy to read "use" before "use timescale" as a verb. After reading the methods, it is suggested to revise as: "We introduce two timescales to describe the net evaporative behavior of a compound from a VCP mixture: evaporation timescale and use timescale." A similar rephrasing is recommended in lines 85-87. In addition, it is recommended that 2.1.5 section heading be written as "Evaporation- and Use-Timescales" or "Evaporation Timescale and Use Timescale" (since these timescales are not hyphenated in the text).**

We updated lines 16-19 to the following:

"*Evaporation of a species from a VCP mixture in the VCPy framework is a function of the compound specific physiochemical properties that govern volatilization and the timescale relevant for product evaporation. We introduce two terms to describe these processes: evaporation timescale and use timescale, respectively.*"

Lines 85-87 have been updated, as well, to be consistent. The heading for section 2.1.5 has been updated.

**lines 24-25: Suggested revision to: "VCPy predicts higher emissions than the NEI for approximately half of the counties, with 5% of all counties having greater than 55% higher emissions."**

Revised.

**line 26: VCPy is introduced as a framework. Thus it is unclear how a framework can report something. Suggest revision to: "Categorially, application of the VCPy yields higher emissions for…"**

We updated this line to:

"*Categorially, application of the VCPy framework yields higher emissions for…*"

**line 27: Suggest to replace "in the methods employed here" with "using the VCPy framework"**

Replaced.

**line 38: Suggest to introduce "O3" here ("tropospheric ozone (O3) and secondary organic aerosol (SOA).") and use throughout. (Appears for the first time in line 470, section 3.5.)**

Introduced.

**lines 49-55: Is there a reason NMVOC and VOC abbreviations introduced here are not "NMVOCs" and "VOCs", whereas VCP abbreviation is introduced as VCPs? The use in all cases is as a plural noun (adjective should be singular), and "VOCs" appears subsequently in the manuscript.**

Updated.

**line 74: Suggest to revise as "biased low"**

Revised.

**lines 89-91: Suggest to revise as: "In addition, we test the sensitivity of predicted emission factors to uncertain parameters, such as evaporation; and use timescales, through Monte Carlo analysis, to evaluate the VCPy inventory using published emission ratios, and to estimate the effective SOA and O3 formation potential of both the complete sector and individual product use categories."**

We updated the sentence to the following:

"*In addition, we test the sensitivity of predicted emission factors to uncertain parameters, such as evaporation timescale and use timescale, through Monte Carlo analysis, evaluate the VCPy inventory using published emission ratios, and estimate the effective SOA and ozone formation potential of both the complete sector and individual product use categories.*"

**line 103: Suggest to revised "organics component" as "organic component" (consistent with previous sentence and subsequent use).**

Revised.

**line 132: Remove "nationally" at the end of the sentence (already established that this is national level). Also, check the use of "national level" throughout. When used as an adjective, sometimes it is hyphenated (e.g., 392, Fig. 7) and sometimes it is not (e.g., line 411, 567). Similarly, recommended changing 3.3 header (line 414) to and Figure S3 caption to: "State- and County-Level …".**

Updated and checked.

**line 153: Suggest to revise as "The organic component is further decomposed into non-evaporative and evaporative organics."**

Updated.

**line 244: "KOA" should be italicized (KOA is preferred throughout (italics only on the variable "K" and not abbreviation "OA", but should at least be consistent with other appearances).**

Updated.

**line 253: The use of "in contrast" is a little confusing since these products are in the same PUC. Suggestion to reword as follows: "As such, these Short Use Personal Care Products are assigned a "Minutes" use timescale. However, it is also assumed that each person bathes once a day and associated Daily Use Personal Care Products are therefore assigned a "Days" use timescale."**

Updated.

**line 359: "compound class" should be "compound classes"**

Updated.

**line 364: "comprise" should be "compose"**

Updated.

**line 555: Suggest to add "O3" before "MIR"**

Added.

**Additional notes:**

In Table S1 and S2, a column header in each table was updated from "NAICS Codes" to "NAICS Product Codes" for additional clarity.